# An Insight into Goat Cheese: The Tales of Artisanal and Industrial Gidotyri Microbiota

**DOI:** 10.3390/microorganisms11010123

**Published:** 2023-01-03

**Authors:** Aikaterini Nelli, Brigkita Venardou, Ioannis Skoufos, Chrysoula (Chrysa) Voidarou, Ilias Lagkouvardos, Athina Tzora

**Affiliations:** Laboratory of Animal Health, Food Hygiene and Quality, Department of Agriculture, University of Ioannina, 47100 Arta, Greece

**Keywords:** goat cheese, Gidotyri, traditional cheese-making, 16S rRNA sequencing, Greek cheese, lactic-acid bacteria, cheese microbiota

## Abstract

The purpose of this study was to determine for the first time the microbiota in artisanal-type and industrial-type Gidotyri cheeses and investigate the influence of the cheese-making practices on their composition using culture-independent techniques. The microbiota present in artisanal with commercial starters (Artisanal_CS, *n* = 15), artisanal with in-house starters (Artisanal_IHS, *n* = 10) and industrial (Ind., *n* = 9) Gidotyri cheese samples were analyzed using a targeted metagenomic approach (16S rRNA gene). The Ind. Gidotyri cheese microbiota were less complex, dominated by the Streptococcaceae family (91%) that was more abundant compared to the artisanal Gidotyri cheeses (*p* < 0.05). Artisanal cheeses were more diverse compositionally with specific bacterial species being prevalent to each subtype. Particularly, *Loigolactobacillus coryniformis* (OTU 175), *Secundilactobacillus malefermentans* (OTU 48), and *Streptococcus parauberis* (OTU 50) were more prevalent in Artisanal_IHS cheeses compared to Artisanal_CS (*p* ≤ 0.001) and Ind. (*p* < 0.01) Gidotyri cheeses. *Carnobacterium maltaromaticum* (OTU 23) and *Enterobacter hormaechei* subsp. *hoffmannii* (OTU 268) were more prevalent in Artisanal_CS cheeses compared to Artisanal_IHS cheeses (*p* < 0.05) and Ind. cheeses (*p* < 0.05). *Hafnia alvei* (OTU 13) and *Acinetobacter colistiniresistens* (OTU 111) tended to be more prevalent in Artisanal_CS compared to the other two cheese groups (*p* < 0.10). In conclusion, higher microbial diversity was observed in the artisanal-type Gidotyri cheeses, with possible bacterial markers specific to each subtype identified with potential application to traceability of the manufacturing processes’ authenticity and cheese quality.

## 1. Introduction

Cheese has constituted an important component of the human diet for millennia, leading to the industrialization of its production. Nevertheless, artisanal cheeses are gaining increasing interest from consumers. As no consensus definition exists, artisanal cheeses are generally considered to be hand-made cheeses produced using traditional cheese-making practices from cow, sheep and/or goat milk (preferably raw) on-farm or in small-scale dairies interlinked to a geographical region and culture [1,2,3]. The traditionally produced artisanal cheeses are characterized by increased microbial diversity associated with superior flavor, aroma and texture, while the standardized manufacturing of industrial cheeses leads to a less complex microbiota in an attempt to improve product safety at the expense of sensorial quality [3].

The composition of the microbial community collectively known as the cheese microbiota has a fundamental role in the production processes of curd formation and ripening, safety, and quality of the final product. Lactic acid bacteria (LABs) are the principal component of this microbial community, while pathogenic and spoilage bacteria such as *Escherichia coli*, *Staphylococcus* spp., *Pseudomonas* spp. are absent or present in low numbers in cheese [4,5,6,7]. Of the LABs, *Lactococcus*, *Streptococcus*, *Lactobacillus* and *Leuconostoc* genera (Starter LABs, SLABs) are responsible for the rapid acidification of milk during the initial fermentation, while *Lactobacillus*, *Pediococcus*, *Leuconostoc* and *Enterococcus* genera (non-starter LABs, NSLABs) influence the development of the organoleptic properties via their proteolytic and lipolytic activities during cheese maturation [8,9]. Apart from their technological importance, several studies have identified dairy-originating LAB strains including ones with the ability to inhibit the growth of pathogens and spoilage microorganisms by acid and bacteriocin production and competitive exclusion, indicative of probiotic potential as well as improved food safety [10,11,12,13,14,15]. Furthermore, the enzymatic processes driven by the cheese microbiota contribute to the production of bioactive components, predominantly peptides and oligosaccharides, with prebiotic, anti-microbial, anti-inflammatory, immunomodulatory, antihypertensive, and intestinal barrier function-enhancing potential, among others [16,17,18,19].

Given the profound importance of cheese microbiota in determining overall cheese quality, significant efforts have been made to identify species variation using traditional and molecular methods. Conventional microbiological techniques have been used in the past to investigate this microbial community; however, these methodologies cannot accurately determine its composition due to their inability to recover non-culturable, less abundant, or rare bacterial taxa [20]. This issue was overcome with the application of high throughput sequencing methods (HTS) that have extended our understanding in cheese microbiology by offering a clearer snapshot of the unique pool of microbes present in each type of cheese [21,22]. HTS has recently been implemented to identify the microbiota of several traditional and PDO cheeses and to uncover the microbial map responsible for the unique sensorial characteristics, thus, contributing to cheese authenticity, marketability, and safety [23,24,25,26,27,28].

Goat cheese is an underestimated food with high nutritional value and a promising potential as a functional food for human nutrition [29,30,31] with Caciotta and Caprino Nicastrese goat cheeses as characteristic examples [1,32]. Within the EU, France followed by Spain and Greece were listed among the 10 top goat cheese producers representing on average 17.3%, 7% and 7.6% of global production for the period 2015–2017, respectively, despite their collective average goat milk production being only 7.9% for the same period [33]. Furthermore, Greece traditionally produces a variety of cheeses from goat milk solely or mixed with sheep milk and/or cow milk [34], a large number (23) of which are registered to eAmbrosia, the EU geographical indications register, under the Protected Designation of Origin (PDO) and Protected Geographical Indication (PGI) labels. Research has mostly focused on identifying the composition of the microbiota, particularly the LAB community, in the industrial or artisanal Greek PDO cheeses, namely Feta, Kefalograviera, Kalathaki Limnou, Arseniko Naxou and Graviera Kritis, using conventional microbiological techniques coupled with molecular tools (Matrix-assisted laser desorption/ionization time-of-flight mass spectrometry, genomic methods) [6,15,28,35,36] or, more recently, HTS [27,37,38,39], while information on other non-PDO traditional cheeses is limited [15,36]. Furthermore, a single study evaluated both artisanal-type and industrial-type galotyri (PDO Greek cheese) using conventional microbiological techniques and reported compositional differences in the LAB community between the two groups [40].

Gidotyri, a traditional non-PDO goat cheese, is produced in a similar manner to Feta cheese [41] from industrial and artisanal dairies and mostly distributed to the local markets of the country. Thus, the aim of the present study was to characterize for the first time the microbiota in artisanal-type and industrial-type Gidotyri cheese and to identify key bacterial species associated with the cheese-making practices using culture-independent 16S rRNA gene HTS.

## 2. Materials and Methods

### 2.1. Collection of Gidotyri Cheese (Goat Cheese) Samples

Thirty-four Gidotyri cheese samples were collected in their original packages (1 kg net weight for industrial Gidotyri cheeses and 500 g net weight for artisanal Gidotyri cheeses) from eight dairy establishments located in the Epirus region production area (Figure 1A) and then transported under refrigeration (4 °C) in no more than two hours, for laboratory analysis. Cheese samples were obtained from six different industrial producers (Ind.) and two artisanal dairies, one using commercial starters (Artisanal_CS) and the other using in-house starter cultures (Artisanal_IHS), with different production procedures (Figure 1B) such as thermalization (63 °C, 15 min for artisanal cheeses) or pasteurization (72 °C, 15 s for industrial cheeses) of milk, type of starter cultures for the acidification procedure, production capacity (small-scale dairies vs industrial dairies). Samples from different production batches were collected from each dairy after 3 months of ripening (milk collection for cheese production took place in July 2022). All cheese samples were collected in November 2022.

### 2.2. DNA Extraction

A 25 g aliquot was sampled from each cheese core once and homogenized in 225 mL of buffered peptone water (LAB M, Bury, Lancashire, UK), using a Stomacher (Laboratory Blender Stomacher 400; Seward, London, UK) for 2 min at 260 rpm. Ten milliliters of the filtered homogenized sample were collected in a 15-mL conical centrifuge tube and high-quality total DNA was extracted using DNeasy PowerFood Microbial kit (Qiagen, Hilden, Germany) according to the manufacturer’s instruction. DNA concentrations were measured using a fluorescence spectrometer (Qubit, Life Technologies, Carlsbad, CA, USA). The samples were stored at −20 °C until analysis.

### 2.3. High Throughput 16S rRNA Sequencing

Aliquots of the obtained DNA of each sample proceeded for the characterization of the microbial diversity through PCR amplification, library preparation and high-throughput sequencing of the V3-V4 region of the 16s RNA gene. The targeted DNA region was amplified using the primers 341F and 806R [42]. The amplicon libraries were prepared using Nextera XT index kit (Illumina Inc., San Diego, CA, USA) and purified using the AMPure XP system (Beckmann Coulter, Krefeld, Germany) according to the manufacturers’ instructions. Sequencing was conducted in a paired-end mode (PE300; only using reads of 275 each) with pooled samples containing 20% (*v/v*) PhiX standard library in the MiSeq Sequencing System (Illumina Inc., San Diego, CA, USA) using the MiSeq Reagent Kit v2 (300-cycles) (Illumina Inc., San Diego, CA, USA) amplifying the 465 bp fragment.

### 2.4. Data Analysis and Bioinformatics

The 16S rRNA gene amplicon data were analyzed and further processed using the “Integrated Microbial Next-generation sequencing” platform based on UPARSE. (IMNGS, www.imngs.org, accessed on 7 July 2022). A de-multiplexing (demultiplexer v3.pl) was performed before the sequences were trimmed by ten nucleotides. Sequences with nucleotides <200 and >600 and expected errors in paired reads >3 were excluded, and samples were screened for chimeras [43]. Operational taxonomic units (OTUs) were clustered at 97% similarity and OTUs with a relative abundance of <0.25% were removed. To generate a graphical overview of the alpha and beta diversity and the microbial composition, taxonomic binning was performed by Rhea using the set of R-scripts described by [44]. For all given results, *p*-values were corrected for multiple comparisons using the Wilcoxon rank-sum and/or Kruskal−Wallis Rank Sum statistical tests, unless stated otherwise. Significant OTUs were then identified at species level by EzBioCloud’s 16S rRNA gene-based ID (www.ezbiocloud.net, accessed on 16 September 2022). Data were visualized using Illustrator CS6 Version 16.0.0 (Adobe Inc., San José, CA, USA).

## 3. Results and Discussion

### 3.1. DNA Sequencing Analysis and Alpha Diversity

A total of 1,399,249 raw paired-end reads were sequenced from the thirty-four Gidotyri cheese samples. After merging quality filtering, chimera removal and normalization, a total of 1,116,188 high-quality sequences were obtained, with an average of 32,829 reads per sample (range 10,425 to 55,131). In total, 222 OTUs were observed.

Alpha diversity metrics, namely Shannon and Simpson diversity, were calculated; however, we considered effective richness as a more accurate measurement of bacterial diversity between the samples, as this index is not affected by sequence depth or normalization steps and takes into account bacterial taxa with a relative abundance over 0.25% for each sample [45]. In this study, both Artisanal_CS and Artisanal_IHS Gidotyri cheeses had increased effective richness compared to Ind. Gidotyri cheeses (*p* < 0.05, Figure 2). Increased microbial diversity has also been observed in previous studies comparing the microbiota of artisanal- and Ind.-type cheeses [27,40,46]. This finding further confirms the assumption that traditional cheese-making practices are directly linked to a more diverse cheese microbial profile.

### 3.2. Beta Diversity

We calculated beta diversity to evaluate the similarity of the microbial profiles between the three Gidotyri cheese groups. Different microbial communities were revealed between all three cheese groups (Permanova *p* < 0.05) demonstrated by the three separated clusters (Figure 3). Similarly, classification of industrial and homemade Feta cheese samples to separate clusters has previously been observed [27]. It is worth noting that, both in the current and the previously mentioned studies, the artisanal cluster was characterized by a higher dispersal of the samples, indicative of the higher variation among the microbial profile of the respective samples. Regarding the increased variability that was evident in the Artisanal_CS cluster, this can be explained by the fact that milk used for each cheese sample was supplied by a different goat farm. Contrarily, Artisanal_IHS cheese samples made from milk produced by a single goat farm formed a tighter cluster. The influence of the raw milk microbiota on the cheese microbial profile has already been extensively reviewed [3,47,48], supporting our observations.

### 3.3. Microbiota Diversity in Gidotyri Cheese Samples

This is the first study that utilized high throughput sequencing to achieve a more in-depth characterization of the microbiota in Artisanal and Industrial Gidotyri cheeses. Four phyla were present in all the samples of the three cheese groups with Firmicutes being predominant. In particular, higher abundance was observed in Ind. Gidotyri cheese samples (99.51%) followed by Artisanal_IHS (99.03%) and Artisanal_CS (88.10%) Gidotyri cheese samples with all being significantly different from each other (*p* < 0.05). Proteobacteria had a higher abundance in Artisanal_CS Gidotyri cheese samples (11.74%) compared to Artisanal_IHS (0.66%) and Ind. Gidotyri (0.43%) cheese samples (*p* > 0.05), while Bacteroidota and Actinobacteriota were present at <0.3% in all three Gidotyri cheese groups. The observed phyla with the predominance of Firmicutes has been observed in a variety of cheese types [4,23,38,46,49,50].

In the present study, we focus on the families with differences in the relative abundance between the three cheese groups; however, the complete list of families identified are presented in Appendix A. Within the Firmicutes phylum, three families were significantly different between the three Gidotyri cheese groups (Figure 4A). Streptococcaceae had a higher abundance in Ind. Gidotyri cheeses (90.96%) compared to the other two cheese groups (73.71% for Artisanal_IHS and 63.85% for Artisanal_CS) (*p* < 0.05). Lactobacillaceae had a higher abundance in Artisanal_IHS (25.25%) compared to the Ind. Gidotyri cheeses (7.63%) (*p* < 0.05). The relative abundance of Lactobacillaceae in Artisanal_CS cheeses (20.48%) was numerically higher than Ind. Gidotyri cheeses (*p* > 0.05) and closer to the Artisanal_IHS. Based on our finding, the microbiota of Ind. Gidotyri cheese is less complex with the Streptococcaceae family representing >90% of the bacterial taxa. Contrarily, Artisanal Gidotyri cheeses include a significant percentage of the Lactobacillaceae family as well. Our findings resemble the ones reported by Samelis and Kakouri [40] that also observed a dominance of members of the Streptococcaceae in industrial galotyri cheeses, while Lactobacillaceae members were more prevalent in artisanal galotyri cheeses. The dominance of Streptococaceae followed by Lactobacillaceae in artisanal goat cheese was also reported in a recent study [51]. Carnobacteriaceae were solely present in Artisanal_CS despite being identified in a single Ind. Gidotyri cheese sample (*p* < 0.05). This family has been considered among the families involved in the acidification of milk during cheese production [3] and has been associated with anti-listerial activity in smear-ripened cheeses [52]. Within the Proteobacteria phylum, two families, namely Enterobacteriaceae and Moraxellaceae, were solely present in the Artisanal_CS despite the former being observed in a single Ind. Gidotyri cheese sample (*p* < 0.05, Figure 4B). These families are commonly found in raw milk and are considered markers of the hygiene conditions during the cheese production process [37,38,53]. There was also a tendency for increased prevalence of Hafniaceae in the Artisanal_CS compared to the other two cheese groups (*p* < 0.10, Figure 4B).

The thirty-five most abundant genera in all three Gidotyri cheese groups are presented in Figure 5, while the complete list of genera identified are given in Appendix A. *Lactococcus* and *Streptococcus* represented the two major genera in all three cheese groups with relative abundance 43.4% and 30.6%, respectively, followed by *Lactiplantibacillus* (8.9%), *Secundilactobacillus* (4.4%) and *Lactobacillus* (2.9%), with the remaining genera being present at <1.6% in accordance with observations from previous studies on different cheese types with variations in the abundances of these genera [23,27,39,46,49,51]. Five genera were found to be significantly different in terms of prevalence (Fisher’s test) between the three cheese groups. *Acinetobacter*, *Carnobacterium* and *Enterobacter* had higher prevalence in Artisanal_CS cheeses (7 out of 15; 10 out of 15; 7 out of 15 samples) compared to Artisanal_IHS cheeses (0 out of 10 for all genera, *p* < 0.05) and Ind. cheeses (0 out of 9; 1 out of 9; 0 out of 9 samples, *p* < 0.05). *Loigolactobacillus* and *Secundilactobacillus* had higher prevalence in Artisanal_IHS cheeses (9 out of 10; 10 out of 10 samples) compared to Artisanal_CS cheeses (2 out of 15; 3 out of 15, *p* < 0.001) and Ind. cheeses (0 out of 9; 2 out of 9 samples, *p* < 0.001). In the present study, the representative species of each genus that dominated in the respective Gidotyri cheese group was identified and will be discussed in the subsequent section.

### 3.4. The Most Prevalent Bacterial Species in Artisanal- and Industrial-Type Gidotyri Cheese

The relative abundance of the bacterial species present in the three Gidotyri cheeses was additionally determined with the 29 most abundant based on their average relative abundance across all cheese samples from all cheese groups being presented in Table 1, along with their respective roles in cheese production and quality. Eight OTUs corresponding to eight different bacterial species were found to be significantly different in terms of prevalence (Fisher’s test) between the three cheese groups. *Loigolactobacillus coryniformis* (OTU 175), *Secundilactobacillus malefermentans* (OTU 48), and *Streptococcus parauberis* (OTU 50) had higher prevalence in Artisanal_IHS cheeses (9 out of 10; 10 out of 10; 9 out of 10 samples) compared to Artisanal_CS cheeses (2 out of 15; 3 out of 15; 2 out of 15 samples, *p* ≤ 0.001) and Ind. cheeses (0 out of 9; 2 out of 9; 1 out of 9 samples, *p* < 0.01). *L. coryniformis* has exhibited antibacterial activity against pathogens and spoilage microorganisms associated with acid, H_2_O_2_ and bacteriocin production [54,55], while *S. malefermentans*, a bacterium exclusively fermenting carbohydrates at low temperature, was recently identified as a core member of sauerkraut (fermented food product) carrying genes encoding for enzymes with significant contribution to the aroma development of the final product [56,57]. *Str. parauberis* is associated with small ruminant mastitis and has been identified as a minor component of cheeses produced with their milk [38,58,59]. Other mastitis-causing pathogens can be found in cheese via contaminated milk [60,61,62,63]. *Carnobacterium maltaromaticum* (OTU 23) and *Enterobacter hormaechei* subsp. *hoffmannii* (OTU 268) had higher prevalence in Artisanal_CS cheeses (10 out of 15; 7 out of 15 samples) compared to Artisanal_IHS cheeses (0 out of 10; 0 out of 10 samples, *p* < 0.05) and Ind. cheeses (1 out of 9; 0 out of 9 samples, *p* < 0.05). *C. maltaromaticum* comprises an important member of the cheese ripening microflora due to its contribution to aroma development, control of spoilage bacteria and anti-listerial bacteriocin production [64,65,66]. The role of *E. hormaechei*, an isolate from several artisanal sheep cheeses, is a bit controversial as it is considered both a poor hygiene indicator of cheese production and a potential contributing bacterium to cheese flavor [61,67,68]. A tendency for higher prevalence of *Hafnia alvei* (OTU 13) and *Acinetobacter colistiniresistens* (OTU 111) was additionally observed in Artisanal_CS compared to the other two cheese groups (*p* < 0.10). *H. alvei* is a frequent member of the microbiota in traditional cheeses possibly related to distinct organoleptic properties while also displaying antibacterial activity against foodborne pathogens [69]. *A. colistiniresistens*, a bacterial species with intrinsic resistance to polymyxins, has not been previously isolated in cheese or other food products [70]. Concerning the two Artisanal cheeses, *Streptococcus thermophilus* (OTU 3) was more prevalent in the Artisanal_IHS (10 out of 10 samples) cheese samples compared to Artisanal_CS (7 out of 15 samples, *p* < 0.05). This bacterial species is a widely used starter culture in cheese and other dairy products with well-known technological properties [71].

As a final point of this study, it is worth mentioning that the type of heat treatment implemented on milk influences the microbial composition of the produced cheese [48,72,73]. Therefore, the more diverse microbiota in artisanal Gidotyri cheeses can probably be attributed to the thermization of the raw milk (63 °C for 15 min) instead of the industrial practice of pasteurization (73 °C for 15 s). We assume that the higher prevalence of bacterial populations used as poor hygiene indicators in Artisanal_CS is likely associated with milk being obtained from multiple sources, namely 15 different goat farms, in contrast with Artisanal_IHS which was solely made from milk of a single goat farm. Based on our findings, specific bacterial species related to artisanal Gidotyri cheeses were detected, indicating the probability of linking traditional cheese-making practices to distinct microbial markers that could be used as traceability models to ensure their authenticity.

**Table 1 microorganisms-11-00123-t001:** The 29 most abundant bacterial species, on average, in Artisanal_CS (Art. CS), Artisanal_IHS (Art. IHS) and Industrial (Ind.) Gidotyri cheese samples with their roles in cheese manufacturing, safety, and quality.

Bacterial Species	Identification Similarity (%) *	Relative Abundance (%)			Spore Forming	Role	Ref.
		Art.CS	Art. IHS	Ind.			
*Lactococcus lactis* subsp. *hordniae*/subsp. *lactis*/*Lactococcus cremoris* subsp. *tructae*	100	42.816	37.530	50.749	no	Starter culture in dairy industry, food safety (production of bacteriocins, nisin)	[74]
*Streptococcus thermophilus*	99.78	20.214	34.533	39.863	no	Traditional starter culture, acidifying activity, food safety, organoleptic properties	[71]
*Lactobacillus delbrueckii* subsp. *bulgaricus*	100	5.588	0.465	0.173	no	Typically found in artisanal cheese, involved in cheese fermentation, production of folates	[75,76]
*Lactiplantibacillus paraplantarum*/*pentosus*/*argentoratensis*/*pingfangensis*	100	10.100	9.966	5.887	no	*Lb. paraplantarum*: found in artisanal cheeses, improving texture (viscosity) by exopolysaccharide production, food safety (production of bacteriocins, paraplantaricin)	[77]
						*Lb. pentosus*: involved in the production of beneficial metabolites (indolepyruvate and pantothenic acid), improves intestinal barrier function (probiotic potential)	[78]
						*Lb. argentoratensis*: capacity to metabolize different carbohydrates, involved in cheese fermentation, riboflavin and folate biosynthesis	[79]
						*Lb. pingfangensis*: isolated from traditional Chinese pickle	[80]
*Hafnia alvei*	100	3.434	0.011	0.228	no	Development of favourable organoleptic properties in cheese, anti-obesity properties in mice	[69]
*Escherichia fergusonii*/*Shigella sonnei*/*Shigella flexneri*	100	1.684	0.022	0.055	no	Poor hygiene indicator of cheese production	[81,82]
*Leuconostoc mesenteroides* subsp. *mesenteroides*/subsp. *cremoris*/subsp. *dextranicum*/subsp. *Jonggajibkimchii*/*Leuconostoc suionicum*	100	2.047	0.000	0.132	no	Flavor-producing starter or adjunct cultures in dairy products	[83]
*Acinetobacter colistiniresistens*	98.88	1.949	0.005	0.031	no	Isolated in human clinical specimens, resistance to polymyxins	[70]
*Secundilactobacillus malefermentans*	99.78	0.921	12.892	0.945	no	Food fermentation at low temperatures (sauerkraut), possible contribution to the aroma development	[56,57]
*Loigolactobacillus coryniformis* subsp. *torquens*	100	1.127	1.285	0.008	no	Isolated from Turkish cheese and goat cheese, antibacterial properties	[54,55]
*Carnobacterium maltaromaticum*	100	2.633	0.007	0.637	no	Psychotropic bacterium isolated in French cheeses, food protection against spoilage and pathogenic bacteria, major role in cheese ripening (favorable aroma)	[64,65,66,84]
*Enterobacter hormaechei* subsp. *hoffmannii*	100	2.773	0.027	0.026	no	Poor hygiene indicator of cheese production, potential contribution to cheese flavour	[61,67,68]
*Acinetobacter albensis*	99.78	0.742	0.009	0.001	no	Isolated in traditional Brazilian cheeses, possible involvement in spoilage and shelf-life (carp fillets)	[7,85]
*Staphylococcus aureus* subsp. *aureus*	99.78	0.487	0.004	0.109	no	Milk contaminant associated with subclinical intramammary infections in ruminants	[86]
*Streptococcus parauberis*	100	0.553	1.642	0.328	no	Isolated in artisanal sheep and goat cheeses and Feta cheese, milk contaminant associated with intramammary infections in small ruminants	[38,58,59]
*Marinilactibacillus psychrotolerans*	99.78	0.322	0.003	0.001	no	Halotolerant, involved in cheese fermentation and ripening, possibly sea salt contaminant	[87]
*Lacticaseibacillus rhamnosus*	99.78	0.323	0.003	0.323	no	Involved in cheese ripening and flavor development, probiotic potential	[88,89]
*Limosilactobacillus fermentum*	99.78	0.286	0.006	0.008	no	Involved in nutritional value, organoleptic and technological properties and preservation of food products	[90]
*Pseudomonas azotoformans*/*lactis*/*carnis*/*paracarnis*/*paralactis*	99.78	0.554	0.013	0.008	no	*Pseud. azotoformans*: pigmented bacteria, causing visual spoilage in dairy foods, case of gray milk and blue pigment formation in cheese	[91]
						*Pseud. lactis*/*paralactis*: involved in spoilage of mozzarella cheese	[92]
						*Pseud. carnis*/*paracarnis*: N/A	-
*Streptococcus caledonicus*	100	0.254	0.000	0.000	no	Isolated from clinical specimens of sheep	[93]
*Exiguobacterium artemiae*	100	0.156	0.000	0.000	no	Isolated in Latin-style cheeses	[94]
*Pediococcus parvulus*	100	0.039	0.620	0.075	no	Antibacterial activity against *Bacillus cereus*	[95]
*Bacillus mycoides*/*cereus*/*pseudomycoides*/*gaemokensis*/*bingmayongensis*/*toyonensis/wiedmannii*/*albus*/*paramycoides*/*proteolyticus*	99.78	0.065	0.000	0.098	yes	*B. mycoides*: cheese spoilage and pink discoloration in Ricotta cheese	[96]
						*B.cereus*: found in dairy products, foodborne pathogen (toxin production), spoilage	[97,98]
						*B. wiedmannii*: isolated from raw milk, cytotoxic member of the *B. cereus* group	[99]
						*Bacillus pseudomycoides*/*gaemokensis*/*bingmayongensis*/*toyonensis*/*wiedmannii*/*albus*/*paramycoides*/*proteolyticus*: N/A	-
*Pseudomonas bubulae*	99.77	0.131	0.001	0.002	no	Isolated from raw refrigerated processed meat of bovine origin	[100]
*Sphingomonas paucimobilis*/*sanguinis*/*yabuuchiae*/*pseudosanguinis*	99.76	0.155	0.461	0.010	no	*Sphingomonas paucimobilis*, *pseudosanguinis*: improving texture (viscosity) by gellan polysaccharide production	[101,102]
						*Sphingomonas yabuuchiae*: improving texture (viscosity) by gellan polysaccharide production, isolated from Irish cheese	[102,103]
						*Sphingomonas sanguinis*: N/A	-
*Shewanella* spp.	99.78	0.053	0.000	0.000	no	Food spoilage, opportunistic human pathogen, goat skin microbiome	[104,105,106]
*Levilactobacillus huananensis*/*lindianensis*	99.55	0.043	0.000	0.000	no	Isolated from traditional Chinese pickle, putative amino acid decarboxylases for biogenic amines production	[107,108]
*Kaistella haifensis*	99.77	0.054	0.274	0.012	no	Isolated from raw milk and Feta cheese from Epirus, lipolytic and proteolytic activity	[38,109,110]
*Weissella thailandensis*	99.55	0.001	0.000	0.079	no	Halotolerant, proteolytic activity, isolated from Mexican cheese	[111]

A color scale is used for the relative abundance, changing from red (absence) to different shades of yellow and reaching green color in response to the increasing values in the corresponding boxes for Artisanal_CS (Art.CS), Artisanal_IHS (Art.IHS) and Industrial (Ind.) Gidotyri cheeses. * The identification at bacterial species level was carried out by BLAST searching of each 16S rRNA sequence on EzBioCloud (https://www.ezbiocloud.net/, accessed on 16 September 2022). N/A: not available.

## 4. Conclusions

In this study, the microbiota of Gidotyri, a traditional Greek goat cheese, was determined for the first time using HTS methodology. Furthermore, the impact of the cheese-making practices, industrial-type versus artisanal-type, was examined. It was clearly demonstrated that artisanal Gidotyri cheeses were characterized by a more complex microbiota with specific families, genera and species linked to each subtype, namely Artisanal_CS and Artisanal_IHS. Contrarily, Ind. Gidotyri cheeses were dominated by the Streptococcaceae family. This initial screening study provides evidence on the presence of possible microbial markers which could serve as authenticity signatures of the manufacturing processes while also possessing distinct technological and beneficial properties that merit further investigation.

## Figures and Tables

**Figure 1 microorganisms-11-00123-f001:**
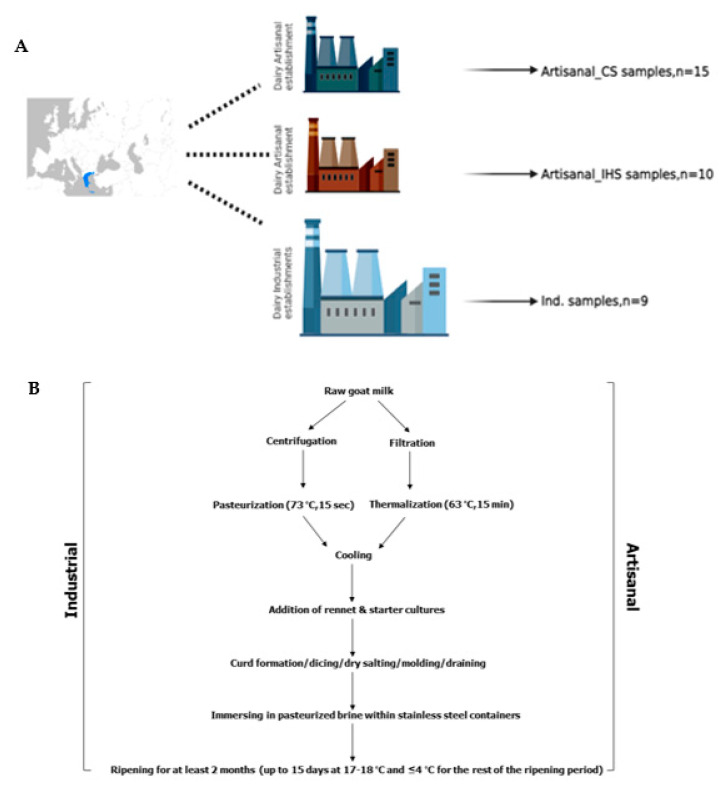
(**A**) A graphical representation of the sampling scheme. For each dairy, samples were collected from different cheese-making lots at the same ripening stage (3 months). The number of Gidotyri cheese samples from each dairy type is presented in the figure. (**B**) The flow chart shows the main production steps of Gidotyri cheese including the differences between industrial and artisanal manufacturing processes. CS: commercial starter cultures; IHS: in-house starter cultures; Ind.: Industrial.

**Figure 2 microorganisms-11-00123-f002:**
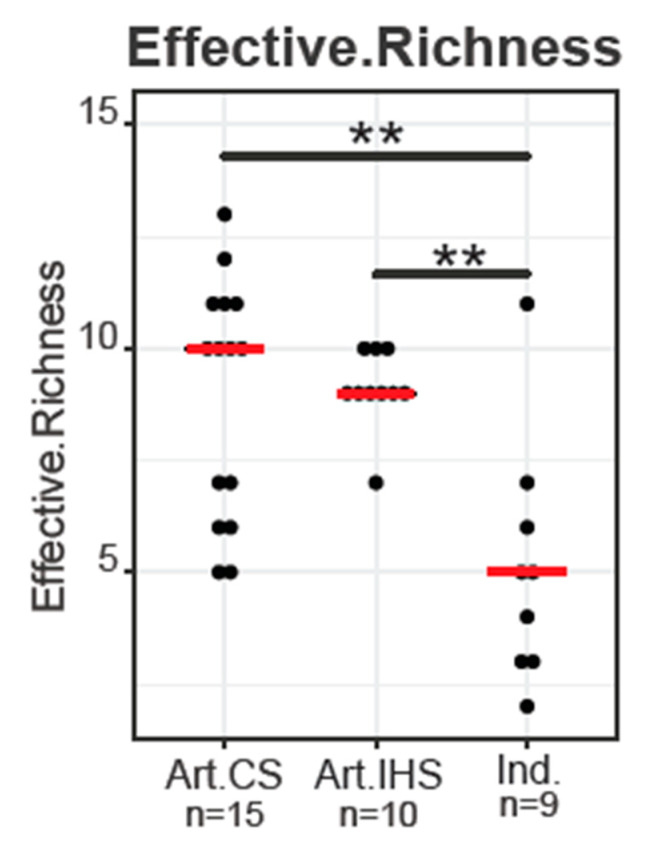
Dot plot of Effective Richness (on the OTU level) between Artisanal_CS (Art. CS), Artisanal_IHS (Art. IHS) and Industrial (Ind.) Gidotyri cheese samples. The red bold lines represent the median. SymbolGidotyri. ** indicates the statistical significance level for the pairwise Mann–Whitney U test after FDR correction (** *p*< 0.01).

**Figure 3 microorganisms-11-00123-f003:**
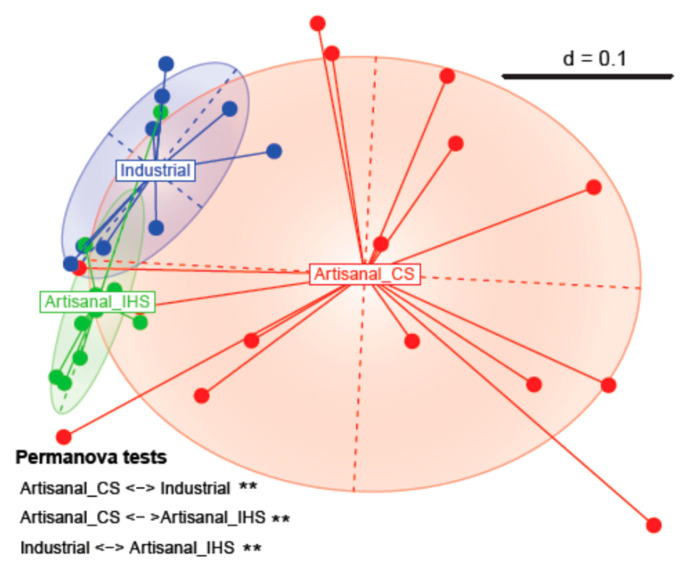
Multidimensional scaling (MDS) plot based on the generalized Unifrac dissimilarity matrix of the microbial profiles from Artisanal_CS, Artisanal_IHS and Ind. Gidotyri cheese samples. SymbolGidotyri. ** indicates the statistical significance level for the pairwise PERMANOVA test after FDR correction (** *p* < 0.01).

**Figure 4 microorganisms-11-00123-f004:**
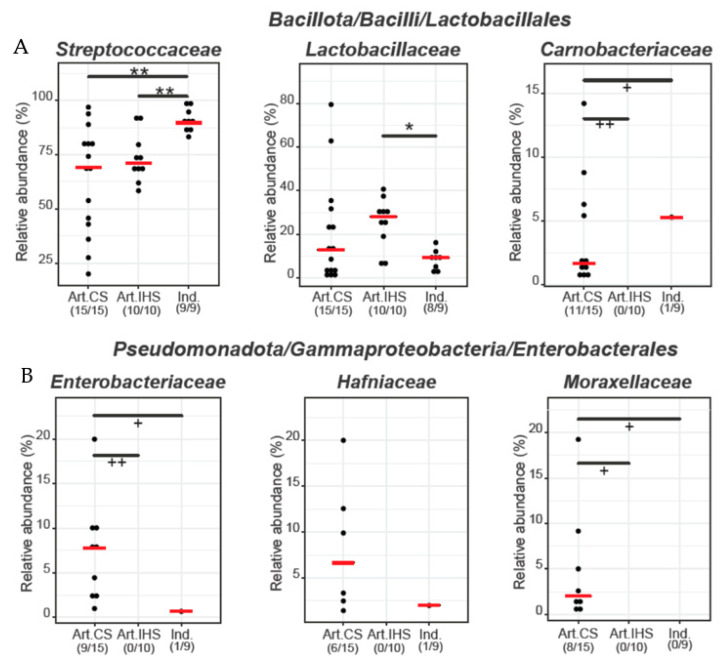
Differential abundance or prevalence of selected bacterial families belonging to Lactobacillales (**A**) and Enterobacterales (**B**) between the Artisanal_CS (Art. CS), Artisanal_IHS (Art. IHS) and Industrial (Ind.) Gidotyri cheeses. The red bold lines represent the median, while symbolGidotyri* indicates statistically significant differences in abundance between cheese groups (Mann–Whitney U statistical test) with the number of stars representing the level of significance (<0.05, * <0.01, ** *p* < 0.01). Symbol (+) indicates statistically significant differences in prevalence between cheese groups (Fisher’s statistical test) with the number of crosses representing the level of significance (+ < 0.05, + + < 0.01).

**Figure 5 microorganisms-11-00123-f005:**
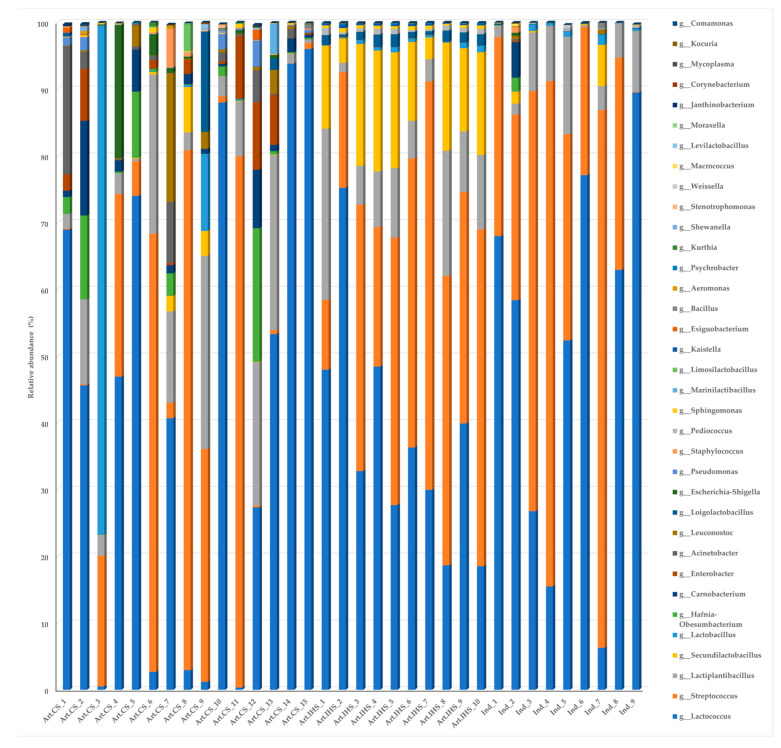
Stacked barplots of the relative abundance of the 35 most abundant genera, on average, in the Artisanal_CS (Art. CS), Artisanal_IHS (Art. IHS) and Industrial (Ind.) Gidotyri cheeses.

## Data Availability

Primary sequencing data were uploaded to ENA public repository with the accession number PRJEB58122.

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
