# Peer review of "An Insight into Goat Cheese: The Tales of Artisanal and Industrial Gidotyri Microbiota"

_microorganisms, 2023, doi:10.3390/microorganisms11010123_

Round 1

Reviewer 1 Report (Previous Reviewer 1)

I thank the authors because they have responded to all my comments satisfactorily. However, there are still a few corrections to be made:

M&M

Please specify the amount of rennet used 

Please specify where the rennet and ferments were purchased

Line 101  refrigeration (4oC).....please correct the temperature

Line 106 see comment above.

Author Response

Comment 1:

Please specify the amount of rennet used. Please specify where the rennet and ferments were purchased

Response 1: Thank you for your comment. The requested information was not available for any of the dairies that produced the evaluated gidotyri cheeses, as it is common practice for private cheesemakers to not reveal their recipes for cheesemaking. All information available has already been included in the Materials and Methods section.

Comment 2:

Line 101  refrigeration (4oC).....please correct the temperature

Response 2: Corrected as requested.

Comment 3:

Line 106 see comment above

Response 3: Corrected as requested.

Reviewer 2 Report (New Reviewer)

The main drawback of this mauscript is the barrage of information about the starter cultures used to make goat cheese. The authors limit themselves to only stating: commercial starter cultures, or in the case of Art IHS- in house starter cultures. Starter cultures may be the reason for the microflora variability shown in this manuscript after cheese ripening.

You should complete the information : what commercial cultures were used (name, manufacturer, country).

In future, when doing such projects, a study of homemade starter cultures should be done by identifying bacterial species using the HTS culture-independent 16S rRNA gene. In this manuscript, we do not know the composition of the home cultures and hence it is uncertain whether small producers in this region add similar primers. It is simpler and faster to test home cultures than the cheeses made from them.

Author Response

Comment 1:

You should complete the information: what commercial cultures were used (name, manufacturer, country).

Response 1: Thank you for your comment. The requested information was not available for any of the dairies that produced the evaluated gidotyri cheeses, as it is common practice for private cheesemakers to not reveal their recipes for cheesemaking. As presented in Table 1 of the manuscript, Lactococcus lactis subsp. hordniae/subsp. lactis/Lactococcus cremoris subsp. tructae and Streptococcus thermophilus were likely the starter cultures for all three cheese groups (Table 1 of the manuscript) as they were the most abundant bacterial species in all cheese samples.

Comment 2:

In future, when doing such projects, a study of homemade starter cultures should be done by identifying bacterial species using the HTS culture-independent 16S rRNA gene. In this manuscript, we do not know the composition of the home cultures and hence it is uncertain whether small producers in this region add similar primers. It is simpler and faster to test home cultures than the cheeses made from them.

Response 2: Thank you for your suggestion. This study was a preliminary investigation of the microbiota in gidotyri, a traditional Greek cheese. Future studies will focus on providing a more in-depth characterization of both starter cultures and cheese microbiota to better identify important bacterial species in cheesemaking linked to specific organoleptic or technological properties.

Reviewer 3 Report (New Reviewer)

Please refer to the attached file with my notes.

Author Response

All recommended edits in the uploaded pdf file were addressed as requested by the reviewer.

This manuscript is a resubmission of an earlier submission. The following is a list of the peer review reports and author responses from that submission.

Round 1

Reviewer 1 Report

GENERAL COMMETS

 The manuscript by Aikaterini Nelli and colleagues is relevant to the topic of the journal, dealing with a current and emerging topic such as cheese microbiota. The article is well written But I have doubts about the description of the experimental plan. However, I believe it can be improved in some sections before publication.

SPECIFICS COMMENTS:

-         Insert the abbreviation of the industrial cheese group (Ind.) throughout the article, e.g. it is missing in the abstract, methods, etc.

-          In the introduction, both when discussing the importance of artisanal and typical cheeses (Line 34-40) and when discussing their impact on human health (Line 71-75), it would be interesting to expand this part by including a recent work done on 5 artisanal milk cheeses in which a "health index" parameter (GHIC) is mentioned precisely.

The manuscript is as follows:  Di Trana, Adriana, et al. "The Quality of Five Natural, Historical Italian Cheeses Produced in Different Months: Gross Composition, Fat-Soluble Vitamins, Fatty Acids, Total Phenols, Antioxidant Capacity, and Health Index." Animals 12.2 (2022): 199.  I suggest inserting it.

2. Materials and Methods

2.1 Collection of gidotyri cheese (goat cheese) samples

-         Line 101. Please describe in more detail and detail the different cheese making procedures, By entering the flowchart of each type of cheese.

-         Also describe the weight of the cheeses, ripening, place of ripening (e.g., on wooden boards), use of salt, use of type of rennet, type of equipment used to process the cheese, exact heat treatment of the milk, and precise description of any cultures used.

-          Line 112: How many repetitions did you use? and which pinto of the cheese was analyzed?  the rind? the core? Specify please

Results & Discussion

Line 194, 3.3 Microbiota diversity in gidotyri cheese samples:

-          To better discuss the results obtained, I recommend that the authors include comparisons with a very recent paper very similar to the present one. Again, this is an investigation of the microbiota of an artisanal raw-milk goat cheese. It would be interesting to compare the results.

The manuscript is as follows: Tilocca, Bruno, et al. "The Functional Characteristics of Goat Cheese Microbiota from a One-Health Perspective." International Journal of Molecular Sciences 23.22 (2022): 14131.

-         Line 305: Did the higher prevalence of bacterial populations used as indicators of poor hygiene in Artisanal_CS generate problems for cheeses during maturation?

-         Furthermore, if your explanation for the presence of these bacteria is the higher numbers of goat farms, then the experimental plan is not robust, as there is a variable that cannot be compared between groups. Probably the presence of these microorganisms is more due to the different heat treatment carried out, which is less drastic and therefore less sufficient to decrease down specific microbial populations?

Author Response

Point 1: Insert the abbreviation of the industrial cheese group (Ind.) throughout the article, e.g. it is missing in the abstract, methods, etc.

Response 1: corrected as requested.

Point 2: In the introduction, both when discussing the importance of artisanal and typical cheeses (Line 34-40) and when discussing their impact on human health (Line 71-75), it would be interesting to expand this part by including a recent work done on 5 artisanal milk cheeses in which a "health index" parameter (GHIC) is mentioned precisely. The manuscript is as follows:  Di Trana, Adriana, et al. "The Quality of Five Natural, Historical Italian Cheeses Produced in Different Months: Gross Composition, Fat-Soluble Vitamins, Fatty Acids, Total Phenols, Antioxidant Capacity, and Health Index." Animals 12.2 (2022): 199.  I suggest inserting it.

Response 2: Thank you for your recommendation. The above-mentioned reference along with another relevant reference (see below) were included in the Introduction section (Line:37,72-73).

Giorgio, D.; Di Trana, A.; Di Napoli, M.A.; Sepe, L.; Cecchini, S.; Rossi, R.; Claps, S. Comparison of cheeses from goats fed 7 forages based on a new health index. J Dairy Sci 2019, 102, 6790-6801, doi:10.3168/jds.2018-15857.

Point 3: Line 101. Please describe in more detail and detail the different cheese making procedures, By entering the flowchart of each type of cheese. Also describe the weight of the cheeses, ripening, place of ripening (e.g., on wooden boards), use of salt, use of type of rennet, type of equipment used to process the cheese, exact heat treatment of the milk, and precise description of any cultures used.

Response 3: Thank you for your comment. We have added a flow chart (Figure 1B) with the production procedure of the different gidotyri cheeses and included the requested information where available in the Materials and Methods section (Line:99-118).

Point 4: Line 112: How many repetitions did you use? and which pinto of the cheese was analyzed?  the rind? the core? Specify please

Response 4: Each cheese sample was analysed once, and the aliquot was obtained from the cheese core as stated in Line 121-122 in the Materials and Methods section.

Point 5: Line 194, To better discuss the results obtained, I recommend that the authors include comparisons with a very recent paper very similar to the present one. Again, this is an investigation of the microbiota of an artisanal raw-milk goat cheese. It would be interesting to compare the results. The manuscript is as follows: Tilocca, Bruno, et al. "The Functional Characteristics of Goat Cheese Microbiota from a One-Health Perspective." International Journal of Molecular Sciences 23.22 (2022): 14131.

Response 5: Thank you for your suggestion. We have included the above-mentioned reference to the relevant parts of the results and discussion section (Line 230-232,251).

Point 6: Line 305: Did the higher prevalence of bacterial populations used as indicators of poor hygiene in Artisanal_CS generate problems for cheeses during maturation?

Response 6: No problems were observed in the maturation process of the Artisanal_CS gidotyri cheeses as well as their organoleptic properties due to the higher prevalence of these bacterial populations.

Point 7: Furthermore, if your explanation for the presence of these bacteria is the higher numbers of goat farms, then the experimental plan is not robust, as there is a variable that cannot be compared between groups. Probably the presence of these microorganisms is more due to the different heat treatment carried out, which is less drastic and therefore less sufficient to decrease down specific microbial populations?

Response 7: Thank you for your comment. The heat treatment for both artisanal cheese types was similar (thermalization at 63oC, 15 min) and therefore is less likely to be the reason for the observed differences in the microbial populations. Regarding the number of goat farms, Artisanal_CS dairy was using milk for multiple farms in the area, in contrast with Artisanal_IHS dairy, which was using milk solely from its own farm for every cheese batch. Based on that, we assumed that the milk obtained from multiple sources was more prone to contamination than if obtained from a single source, explaining the observed higher prevalence of bacterial populations associated with poor hygiene. We have edited the relevant part of the Results and Discussion section accordingly (Line 314-325).

Reviewer 2 Report

This work aims to be a description of 3 types of goat cheese: artisanal with commercial starter, artisanal with natural starter and industrial  by pasteurised milk. There are few existing works on this subject, and this research could be of interest but in my opinion it was not carried on  rigorously. I’ll list my concerning’s

There are important missing info in material and methods: i) It is not clear how natural starters are produced ii) it is not clear which are the bacterial species accounting for the commercial starters (what about the company producing this starter?) and which one for the natural starter, iii) it is unknown when  the cheese samplings were made (all the same day? Different week? which kind of season?)

In addition there is no quantitative information: this paper is focused only on microbial aspects of this goat cheese so I’d  expect to have not only qualitative but also quantitative data, e.g. bacterial counts on plates;  in addition plate counts would have allowed a confirming of the Illumina data by isolation and 16S sequencing of bacterial colonies: it is not uncommon that the 16S sequencing of bacterial isolates reveal that some species appearing in illumina sequencing are not really viable.

 Furthermore, the data were not processed adequately at a statistical level

More in detail:

Section 2.1

It seems that you collected 15 samples from one CS dairy , 10 samples from one IHS dairy  and 9 samples from 6 different industrial diaries. they were representative of the same day? Same week? Same month? It is important to know when you collected the samples to better understand  the meaning of the results about diversity analysis. It is known that season could affect bacterial diversity in artisanal dairy production

Regarding industrial production: 9 samples for 6 dairies? that means for some dairies you have only one sample: this is not robust from a statistical  point of view

Please lines 103 – 104 change in “Cheese samples were obtained from six industrial producers (IND) and two artisanal dairies, one using commercial starter (CS) and one using natural starter cultures (NSC) produced …. (describe how the NSC is produced )

Change IHS with NSC because IHS is commonly used as the abbreviation  of Christ name.  

Please upload your sequencing raw data on a public database (e.g. SRA of NCBI )

Lines 154-155: you could not analyse OTUs data expressed as relative abundance with parametric or non-parametric statistical tools (Wilcoxon rank-sum and/or Kruskal−Wallis Rank Sum statistical tests or Fisher test at line 252 and so on … ) you have to do a log transformation and use proper tools. Just for instance: differential abundance analysis could be analysed by edgeR, metagenomeSeq, DESeq2, analysis of compositions of microbiomes (called ANCOM), a zero-inflated beta model (ZIBSeq), a zero-inflated generalized Dirichlet-multinomial model (ZIGDM), a count regression for correlated observations with a beta-binomial model (corncob). I suggest to analyse your data with a qiime2 or other pipeline (in R there are good tools for statistical OTUs analysis) and to change material and methods according the statistical pipeline used.

Please write in italic the taxa bacterial name

Fig 5 and Table 1 what do you mean for “most abundant genera”? please change figure and table captions according to the definition

Table 1 please list the bacterial species from the more to the less abundant considering OTUs numbers.

There are some species present with an abundance lower than 0.1 % for sure they are not to be considered among the” most abundant” bacterial species. I suggest to consider them as species present in traces. Eg. Bacillus spp, Shewanella spp, levilactobacilus spp, weissella thailandensis are never present in relative abundance higher than 0.1% . Please erase their rows and  add a new row, call it as “others/unassigned” and cluster all the other taxa present in traces or not assigned in the new row

Please consider the taxa and do not list all the species belonging to the genus e.g Lactiplantibacillus paraplan-tarum/ pentosus/argentora-tensis/pingfangensis I suggest to summarise them as Lactiplantibacillus spp.  and so on …

Again  Table 1 please delete identification similarity column

Reviewer 3 Report

The topic of the presented work is very current and very interestingly processed. Nevertheless, I have a few comments:

1. Methodology: L 100 – it is advisable to add the exact parameters of the transport of samples to the laboratory (degrees, duration of transport), which can also partially affect the microbiota of cheeses.

2. Results and discussion: L 302 to L 310 – these claims need to be supported by other studies and discussed more.

Author Response

Point 1: Methodology: L 100 – it is advisable to add the exact parameters of the transport of samples to the laboratory (degrees, duration of transport), which can also partially affect the microbiota of cheeses.

Response 1: corrected as requested (Line 102).

Point 2: Results and discussion: L 302 to L 310 – these claims need to be supported by other studies and discussed more.

Response 2: Thank you for your comment. We have edited the above-mentioned part accordingly and added references, where relevant, in the Results and Discussion section (Line 314-325).

Round 2

Reviewer 1 Report

I thank the authors because they scrupulously followed my suggestions and updated the manuscript correctly according to all my comments.

In my opinion, the paper can be published and I congratulate the authors.